# Modeling and Disturbance Compensation Sliding Mode Control for Solar Array Drive Assembly System

Ji Liang [1,2,3], Hongguang Jia [1,3], Mao-Sheng Chen [1,3], Ling-Bo Kong [3], Huiying Hu [3] and Lihong Guo [1,*]

1    Changchun Institute of Optics, Fine Mechanics and Physics, Chinese Academy of Sciences, Changchun 130033, China; liangji19@mails.ucas.ac.cn (J.L.)
2    University of Chinese Academy of Sciences, Beijing 100049, China
3    Chang Guang Satellite Technology Co., Ltd., Changchun 130102, China
*    Correspondence: guolh@ciomp.ac.cn

**Abstract:** In this study, a dynamic model of a solar array drive system that includes a pair of flexible solar arrays with a central rigid shaft and a permanent-magnet synchronous motor (PMSM) was developed, and a disturbance compensation sliding mode control (DCSMC) strategy was proposed to realize the speed smoothing and vibration suppression control of the system. The continuous nonlinear dynamic equation of the system was derived from Hamilton's principle, and its linearized form was combined with the boundary conditions to obtain its natural frequency and global mode. The design of the DCSMC strategy was based on the solar array drive assembly (SADA) electromechanical dynamics model of the PMSM direct drive. An extended state observer (ESO) was used to estimate any system disturbances, and the signal was fed forward to sliding mode control (SMC) based on the varying gain saturation reaching law (VGSRL). To verify the validity of the model, its results were compared with those obtained using commercial finite element software. The numerical results showed that the SADA system with the DCSMC strategy outperformed the traditional proportional–integral (PI) control and SMC systems.

**Keywords:** solar array drive assembly; permanent-magnet synchronous motor; sliding mode control; global mode

## 1. Introduction

Solar arrays need to be closely oriented toward the Sun to improve the energy-acquisition efficiency of satellites in orbit. Regarding a body-mounted solar array, regular attitude maneuvers are required to adjust its direction [1,2] and thereby shorten the effective working time of onboard devices, such as remote sensing cameras and antennas. Solar array drive systems (SADSs) have been widely used to meet the increasing energy demand of power devices on small satellites and thereby meet the requirements for both satellite energy acquisition and the working time of payload devices. These systems generally consist of a solar array and solar array drive assembly (SADA). The development of the high-precision and high-resolution remote sensing satellite industry has resulted in higher requirements for the pointing accuracy of satellite systems; SADA drive disturbances prominently affect this accuracy. Therefore, it is essential to establish a SADA dynamic model, including the sailboard, conduct research on highly smooth controls for drive systems, and reduce the impact of the system disturbances experienced by satellite platforms. With the support of SADAs, solar arrays on satellites rotate continuously relative to the central satellite to ensure that their normal lines are oriented toward the Sun. A SADA mainly comprises a drive source (motor), drive controller, drive output device, and other components. Harmonic torque, cogging torque, and friction torque are types of nonlinear disturbances that affect motor operation and result in the large rotation of the load, which is the flexible solar array, creating a rigid–flexible coupling effect between the rigid shaft

and array. These factors lead to complexities in the dynamic characteristics of the driving process and changes in disturbance.

A rigid shaft and a flexible solar array with flexible hinges (SSH) can generally be considered a typical rigid–flexible coupling structure or a central rigid-body flexible beam model with flexible links (RFF). Since it is a flexible structure, a solar array has infinite degrees of freedom. To analyze the nonlinear dynamic characteristics of a system or design a controller based on its dynamic model, it is necessary to discretize its continuous displacement and obtain a dynamic model with limited degrees of freedom.

The finite element method (FEM) can discretize rigid–flexible structures with complex shapes and achieve high precision when there are sufficient elements. This method has been used by many researchers to analyze and study dynamic characteristics. Yang et al. [3] adopted the FEM to obtain a finite-dimensional model of an Euler–Bernoulli beam that can account for a large range of rotational motion. This model considered the coupling between the transverse and axial vibrations, the elastic deformation of the beam, and the coupling of a large range of rotational motion. However, the form of this model was complex, and further simplification was needed to facilitate control. Gasbarri et al. [4] used the FEM to establish a detailed structural model of a satellite and obtain its normal mode and natural frequency. Li et al. [5] proposed a new parallel computing method to solve the differential-algebraic equations that describe the multibody system of a grid reflector, reduced the dimension of the linear equations generated by the FEM using the absolute node coordinate formula, and improved the computational efficiency of the dynamic equations. Generally, the results obtained by the FEM are not analytical, and it is difficult to directly use them to analyze the nonlinear dynamic characteristics of a system or design its controller.

The modal method uses an analytical modal function to discretize the dynamic equation of a continuous system and obtain a finite-dimensional modal equation through modal truncation. Compared with the performance of the finite element method, this method can significantly improve the computational efficiency involved in solving the dynamic equation of a system and provide the basis for the analysis of the nonlinear dynamic characteristics of a system and the design of a controller. Gao et al. [6] used the hypothetical mode method to deduce an n-order modal dynamic equation of a flexible beam disturbed by unknown spatiotemporal changes in a tangent coordinate system based on the Lagrange equation. However, this model only considered the flexible beam and ignored the rigid–flexible coupling effect of the system. Celentano et al. [7] used a method based on the assumed modal method to obtain the analytical dynamic model of an entire robot. In this method, an appropriate linear combination of the modes of each link was used as the basis function to evaluate the deflections and reduce the number of items involved in the design of the model. Then, an iterative interconnection algorithm was employed to integrate each deflection and obtain the model. When using the assumed modal method to deal with composite structures, it is difficult to simultaneously satisfy all the geometric and force boundary conditions because of the interrelation between the various components. Additionally, the rationality and accuracy of this method are questionable. Thus, while the modal synthesis method can establish a dynamic model of composite structures, the results obtained using this method are inaccurate and too complex [8].

To overcome these shortcomings, the global modal method (GMM), which uses only one group of time coordinates to discretize the dynamic model, was proposed. Compared with the dynamic model obtained by other methods, that obtained by GMM has low dimensions and high accuracy [9]. He et al. [10] simplified a large flexible spacecraft as a central rigid body, hinged it with two groups of multi-panel structures, and obtained discrete dynamic equations using the GMM. Wu et al. [11] established an analytical dynamic model of a ring truss antenna structure using the GMM and compared the vibration mode and frequency obtained by this method with those obtained by commercial finite element software to verify the reliability of this method. Then, they analyzed the influence of geometric and physical parameters on the natural frequency of the structure. Wei et al. [12] obtained a reduced-order analytical dynamic model of a single flexible-link flexible-joint

(SFF) manipulator using the GMM. Cammarata [13] used the frame provided by the finite element floating reference frame formula (FEM-FRF) and proposed a reduction method for planar flexible mechanisms based on the GMM.

In recent years, many studies have focused on methods for controlling the direction in which solar arrays are oriented toward the Sun. These methods can be divided into two types based on whether a stepper motor with a reducer drive or a permanent-magnet synchronous motor (PMSM) with a direct drive is used. A stepper motor with a reducer is widely used as the driving source for SADA because of its low cost and simple control [14–17]. Chen et al. [14] modeled a SADA system driven by a two-phase hybrid stepper motor and verified its accuracy in driving rigid loads through ground experiments. Cao et al. [15] adopted a strategy combining sliding mode current compensation and output shaping technology to compensate for the torque during SADA driving and achieved good simulation results. Sattar et al. [16] analyzed the disturbance characteristics of a stepper motor driving a SADA and conducted experiments on a piezoelectric testing platform to verify the applicability of different mathematical methods. Zhang et al. [17] used a magnetorheological actuator to suppress the vibration of a solar array, significantly reducing the disturbance torque. The harmonic torque generated by a stepping motor drive is large, and the structure of the reduction mechanism is relatively bulky, making it easy to introduce other disturbances during operation. However, there are some shortcomings of using a stepping motor to drive a flexible solar array.

Recently, researchers have proposed an active control scheme using a PMSM as the driving source. This scheme directly drives a flexible load using a driving mechanism without a reducer. Zhou et al. [18] used an adaptive robust controller in the speed loop of a PMSM to ensure that the system was both uniformly bounded and uniformly ultimately bounded to offset the uncertainty of the system. The effectiveness of the system was verified through numerical simulation. Guo et al. [19] adopted a PMSM as the driving unit, applied a proportional–integral (PI) controller combined with a phase compensation strategy to realize the active control of a solar array, and conducted ground and in-orbit experiments to verify the superiority of the PMSM drive. These studies showed the effectiveness of the PMSM direct-drive scheme in achieving the high-stability driving of solar arrays; therefore, this scheme was adopted in this study.

Regarding a SADA system, the effect of coupling torque on fluctuations in the rotational speed can be regarded as a nonlinear disturbance. Simultaneously, the driving process of a SADA system is affected by friction torque, motor cogging torque, and other nonlinear disturbances. Such a system exhibits multivariable, strong coupling and nonlinear characteristics. The accuracy of traditional linear controllers, such as PI controllers, depends on the system model, which is easily affected by external interference and internal parameter changes and can only reach a control accuracy within a certain range. This makes it difficult to meet the control requirements of a SADA system and results in the control system potentially deviating from the expected goal [20]. Sliding mode control (SMC) has become the focus of research on PMSM driving complex loads because of its low model requirements and high robustness to external disturbances [21–23].

In practical applications, SMC may create high-frequency chattering in a system owing to the time delay in the switching control. Researchers have employed several methods, such as reaching laws [24], the high-order sliding mode method [25], and the nonsingular terminal sliding mode [26], to suppress this phenomenon. Among these, the approach law design method can more directly affect the approach process and is effective at suppressing chattering.

Another problem with SMC is that it is difficult to simultaneously satisfy requirements for the strong robustness and high stability of the system. Specifically, the robustness of SMC is based on the switch gain set in the controller being sufficiently large to offset the interference. However, in practice, it is difficult to determine not only the upper and lower bounds of external interference but also the effect of the switch function in the controller, which significantly increases the amplitude of chattering and affects

the stability of the system operation. Adding feedforward compensation based on a disturbance observer (DOB) to SMC can improve the disturbance rejection capability without deteriorating the control performance of the system and can effectively resolve the contradiction between the high stability and strong robustness of the system. Yang et al. [27] combined a DOB based on iterative learning with a fast-integration terminal SMC law. The experimental results showed that this strategy could ensure a good speed-tracking performance of the PMSM drive system and effectively suppress periodic disturbances. Lu et al. [28] designed a second-order nonsingular terminal sliding mode load observer for a PMSM with a low-speed and high-torque drive load. The simulation results showed that this method improved the robustness and disturbance resistance of the system. Xu et al. [29] used an extended state observer (ESO) approach based on SMC called SMESO to estimate the total disturbance of a system and input the signal into an FTSMC controller for feedforward compensation to improve the performance of the system.

In this study, we propose (1) an analytical dynamic model of a rigid shaft and flexible solar array with a flexible hinge (SSH) that was obtained using the GMM and (2) a disturbance compensation sliding mode controller (DCSMC), which combines a varying gain saturation reaching law (VGSRL) and an ESO. Section 2 details the established dynamic model of the driving shaft and solar arrays with a hinge, and Section 3 explains the design of the DCSMC based on the mathematical model of the PMSM and rigid–flexible coupling load and the verification of the stability of the controller. Section 4 describes the verification of the validity of the SSH model through a comparison with the results of commercial finite element software. Subsequently, control effects under different conditions are also discussed. Finally, Section 5 summarizes the main points of the study.

## 2. Dynamic Model of a Driving Shaft and Solar Arrays with Flexible Hinges

### 2.1. Descriptions and Assumptions for the Proposed Model

The model of the shaft and solar arrays with a rigid platform and flexible hinges (SSH) is illustrated in Figure 1. The PMSM model is represented by the motor symbol. The motor drives the rigid shaft directly without a reduction mechanism. To simplify the results of the study, the following assumptions have been made:

1. Considering that the mass of the satellite is significantly larger than that of the solar array when the solar array is driven at a low speed, the central rigid body is regarded as a fixed reference body.
2. The flexible hinge has been simplified as a hinge with an additional torsion spring, and the mass, size, damping, and friction of the torsion spring have been ignored. The solar arrays have been fully extended, and the hinge has been locked.
3. The rotating shaft is regarded as a rigid body, and the ratio of the length to the width of the solar arrays is sufficiently large to ignore the effect of transverse shear when elastic displacement occurs. The solar array is applicable to the theoretical Euler–Bernoulli beam model.
4. The permanent magnet in the motor is ideal and ignores the effects of magnetic saturation, hysteresis, and eddy currents. The motor's magnetic circuit is linear, and the stator's winding current generates only a sinusoidally distributed magnetic potential in the air gap, ignoring the high-order harmonic magnetic potential in the magnetic field.

Based on assumption 3, the central rigid body, together with the internal drive mechanism, is regarded as the reference body without considering the overall attitude motion. As shown in Figure 2, the reference coordinate system is located at the center of the satellite platform, the body coordinate system of the solar panel is located on the drive axis, and the $Z_0$- and $Z_1$-axes are along the rotation axis. The coordinate system $O_1 - X_1 Y_1 Z_1$ rotates around the $Z_1$-axis with the rotation of the solar array. Regarding the variables, $\theta_s$ is the angular displacement; $S_{1,2}$ is the flexure hinge; $\theta_{k1,k2}$ are the flexure-hinge torsion angles;

and $L$ and $b$ are the length and width of the solar array, respectively. The cross-sectional geometry of the coordinate system is shown in Figure 2, where point $P_0$ is any point on the system, $P$ represents the position of $P_0$ after lateral displacement, and $\tau_s$ is the driving torque acting on the rotating shaft.

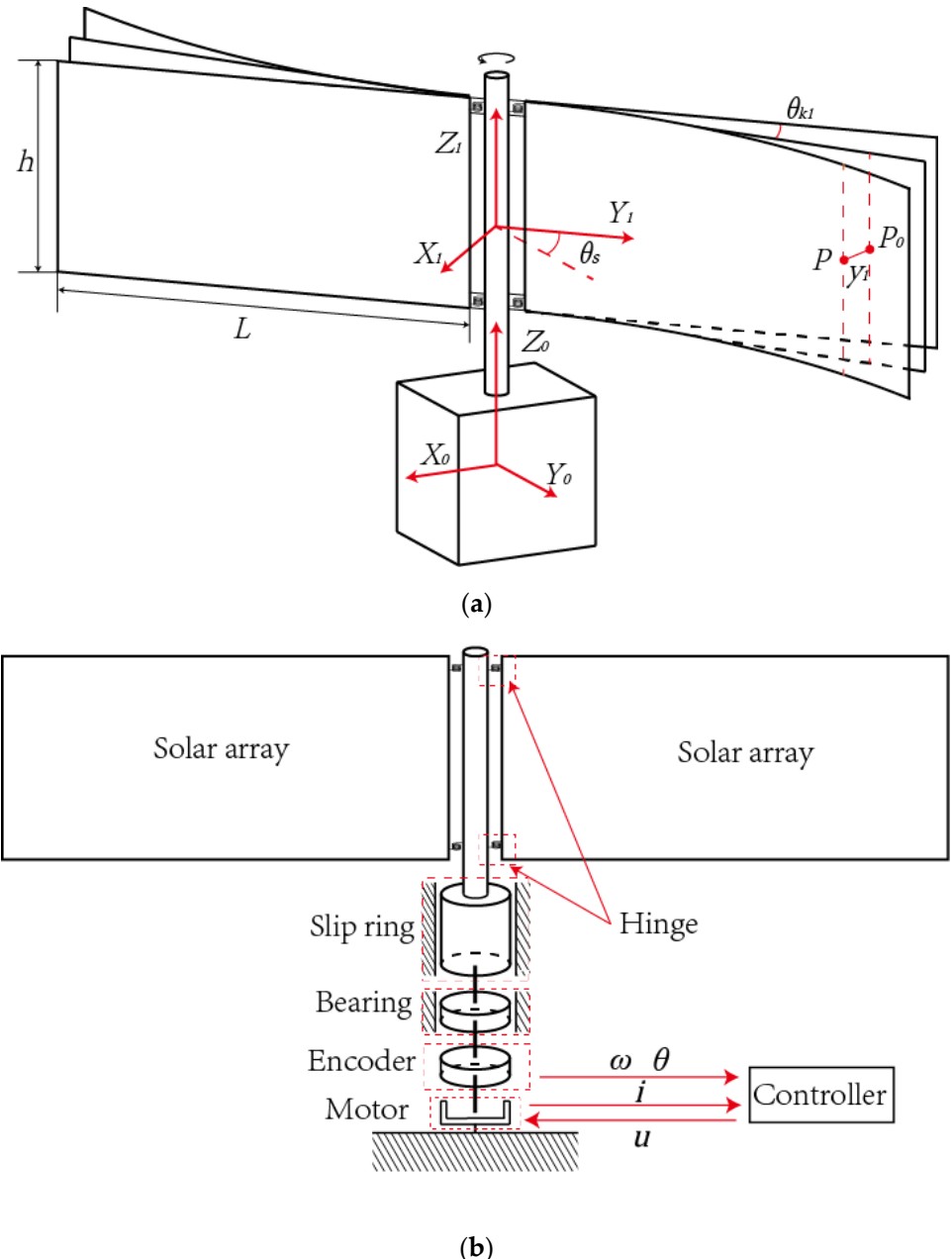

**(a)**

**(b)**

**Figure 1.** (**a**) Coordinate system of satellite and solar array; (**b**) the model of the solar array drive assembly.

### 2.2. Dynamic Model of the Load System

The kinetic energy of the system is:

$$T = \frac{1}{2}J_s\dot{\theta}_s^2 + \frac{1}{2}\int_r^{r+L}\rho\left(x\left(\dot{\theta}_s + \dot{\theta}_{k1}\right) + \dot{y}_1\right)^2 dx + \frac{1}{2}\int_{-r}^{-r-L}\rho(x(\dot{\theta}_s + \dot{\theta}_{k2}) + \dot{y}_2)^2 dx, \quad (1)$$

where $\rho$ is the linear density of the solar array, $y_i$ is the elastic displacement of a point on the i-th solar array, and $x$ is the abscissa of the point.

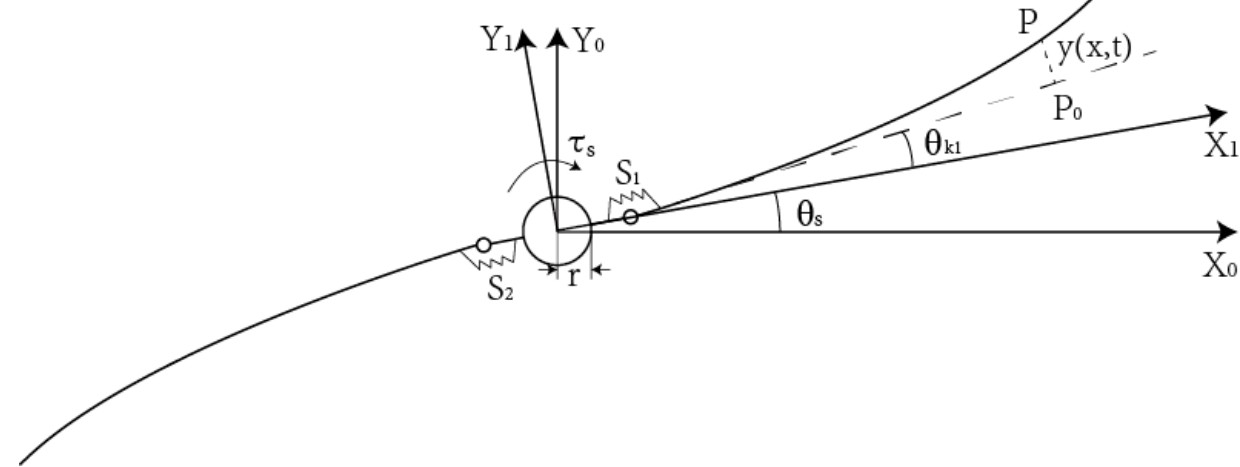

**Figure 2.** Geometric model of rigid shaft and flexible solar arrays with flexible hinge.

The system's potential energy is:

$$V = \frac{1}{2} \int_r^{r+L} EI_z \left( \frac{\partial^2 y_1}{\partial x^2} \right) dx + \frac{1}{2} \int_{-r}^{-r-L} EI_z \left( \frac{\partial^2 y_2}{\partial x^2} \right) dx + \frac{1}{2} k_k \theta_{k_1}^2 + \frac{1}{2} k_k \theta_{k_2}^2, \tag{2}$$

where $E$ is Young's modulus; $I_z$ is the cross-sectional moment of inertia of the solar array. The work of the motor torque acting on the shaft is:

$$W = \tau_s \theta_s, \tag{3}$$

The system dynamics equation in continuous form is obtained from the Hamilton variation principle:

$$J_t \ddot{\theta}_s + J_L \left( \ddot{\theta}_{k1} + \ddot{\theta}_{k2} \right) + \rho \int_r^{r+L} x \ddot{y}_1 dx + \rho \int_{-r-L}^{-r} x \ddot{y}_2 dx = \tau_s, \tag{4}$$

$$EI_z y_1'''' + \rho \left( x \left( \ddot{\theta}_s + \ddot{\theta}_{k1} \right) + y_1 \right) = 0, \tag{5}$$

$$EI_z y_2''''' + \rho \left( x \left( \ddot{\theta}_s + \ddot{\theta}_{k2} \right) + y_2 \right) = 0, \tag{6}$$

where $J_t = J_s + 2J_L$, $J_L = 2\rho \int_r^{r+L} x^2 dx$.

The corresponding boundary conditions are:

$$\begin{cases} y_1(r,t) = 0 \\ y_1''(r+L,t) = 0, \quad y_1'''(r+L,t) = 0 \end{cases}, \tag{7}$$

$$\begin{cases} y_2(-r,t) = 0 \\ y_2''(-r-L,t) = 0, \quad y_2'''(-r-L,t) = 0' \end{cases} \tag{8}$$

According to the assumption, the connection between rigid and flexible bodies is regarded as a torsion spring, and the nonlinear expression of the transmitted torque is:

$$M_i^T = c\dot{\theta}_{ki} + k_L \theta_{ki} + k_N \theta_{ki}^3 + \mu Sign\dot{\theta}_{ki}, \quad k = 1,2, \tag{9}$$

To simplify the analysis, its approximate linear form is usually adopted:

$$M_i^T = k_L \theta_{ki} \quad (k = 1,2), \tag{10}$$

According to the matching conditions of displacement and the rotation angle, force, and moment between rigid and flexible bodies, the matching conditions of the hinge can be obtained as follows:

$$\begin{cases} y_1'(r,t) = \theta_{k1} \\ EI_z y_1''(r,t) = k_L \theta_{k1} \end{cases}, \tag{11}$$

$$\begin{cases} y_1'(-r,t) = \theta_{k2} \\ EI_z y_2''(-r,t) = k_L \theta_{k2} \end{cases}, \tag{12}$$

In order to obtain the analytical mode of the system, $\tau_s$ in Equation (4) is set to zero. The elastic displacement is expressed as:

$$y_i(x,t) = \varphi_i(x)sin\omega t, \ \ i = 1,2, \tag{13}$$

where $\varphi_i(x)$ is the modal function of the $i$-th solar array, and $\omega$ is the circular frequency of the system.

The literature [30] points out that when the external force drives the rigid–flexible coupling structure, the motion generated by the rigid body can be divided into two parts: the first is the large-scale motion generated by the overall system under the external force. The second is the vibration between the rigid body and the flexible body, which involves coupling and synchronization. The angular displacement of the system can be expressed as:

$$\theta_s = \theta_{sr} + \theta_{sv}, \tag{14}$$

where $\theta_{sr}$ and $\theta_{sv}$ represent rigid-body motion and vibration, respectively. In the modal analysis, the external force is set to zero, so the large-scale rigid-body motion $\theta_{sr}$ is zero. The vibration of the rigid body and flexible solar array are coupled and synchronized, so $\theta_{sv}$ is expressed in the same form as elastic displacement. Therefore, Equation (14) can be expressed as:

$$\theta_s = \theta_{sv} = \theta_{s0}sin\omega t, \tag{15}$$

Substituting Equation (15) into (4) and combining (13) results in the following expression of $\theta_{s0}$:

$$\theta_{s0} = -\Gamma_0 / J_t, \tag{16}$$

where $\Gamma_0$ is a notation for simplifying the expression and expressed as:

$$\Gamma_0 = J_L(\theta_{k1} + \theta_{k2}) + \rho \int_r^{r+L} x\varphi_1 dx + \rho \int_{-r-L}^{-r} x\varphi_2 dx, \tag{17}$$

Substituting Equations (14) and (15) into (5) and (6) and combining (13) and (16), the ordinary differential equation of $\varphi_i(x)$ is obtained as follows:

$$\varphi_i^{(4)}(x) - \lambda^4 \varphi_i(x) = -\lambda^4 \Gamma_0 x / J_t, \ \ i = 1,2, \tag{18}$$

where

$$\lambda^4 = \frac{\omega^2 \rho}{EI_z}, \tag{19}$$

The solution of the ordinary differential equation can be expressed as:

$$\varphi_i(x) = \varphi_{ig}(x) + \varphi_{ip}(x), \tag{20}$$

where $\varphi_{ig}(x)$ and $\varphi_{ip}(x)$ are the general and particular solutions of the differential equation, which can be, respectively, written as:

$$\varphi_{ig}(x) = C_{i1}cosh\lambda x + C_{i2}sinh\lambda x + C_{i3}cos\lambda x + C_{i4}sin\lambda x, \tag{21}$$

$$\varphi_{ip}(x) = C_{i5}x + C_{i6}, \tag{22}$$

where $C_{ij}(\lambda)$ ($i = 1, 2$, $j = 1 \sim 6$) are the unknown coefficients.

Substituting Equation (21) into (18), the coefficients $C_{i5}$ and $C_{i6}$ are obtained:

$$C_{i5} = \Gamma_{0g}/J_t, \, C_{i6} = 0, \tag{23}$$

where $\Gamma_{0g}$ is a notation for simplifying expression and expressed as:

$$\Gamma_{0g} = J_L(\theta_{k1} + \theta_{k2}) + \rho \int_r^{r+L} x\varphi_{1g}dx + \rho \int_{-r-L}^{-r} x\varphi_{2g}dx, \tag{24}$$

where $\theta_{ki}$ ($i = 1, 2$) are the unknown coefficients.

Substituting Equation (21) into (24), $\Gamma_{0g}$ is written as follows:

$$\begin{aligned}\Gamma_{0g} = C_{11}\xi_{11}(\lambda) \quad &+C_{12}\xi_{12}(\lambda) + C_{13}\xi_{13}(\lambda) + C_{14}\xi_{14}(\lambda) + C_{15}\xi_{15}(\lambda) \\ &+C_{16}\xi_{16}(\lambda) + C_{17}\xi_{17}(\lambda) + C_{18}\xi_{18}(\lambda) + J_L\theta_{k1} + J_L\theta_{k2},\end{aligned} \tag{25}$$

where the terms $\xi_{ij}(\lambda)$ ($i = 1, 2$, $j = 1 \sim 4$) are listed in Appendix **??**.

The solution of Equation (17) is obtained as:

$$\varphi_i(x) = C_{i1}B_{i1}(x) + C_{i1}B_{i1}(x) + C_{i1}B_{i1}(x) + C_{i1}B_{i1}(x) + b_i(x), \tag{26}$$

where the terms $B_{ij}(\lambda)$ ($i = 1, 2$, $j = 1 \sim 4$) and $b_i(x)$ ($i = 1, 2$) are listed in Appendix **??**.

In order to obtain the values of constants $C_{ij}(\lambda)$ and $\theta_{ki}$, it is necessary to combine Equation (26) with boundary condition Equations (7) and (8) and hinge matching condition Equations (11) and (12). The characteristic equation of the system is obtained as follows:

$$\boldsymbol{H}(\omega)\boldsymbol{\Psi} = 0, \tag{27}$$

where the detailed expression of $H(\omega)$ is given in Appendix **??**. $\boldsymbol{\psi}$ is the undetermined coefficient vector, which is expressed as:

$$\boldsymbol{\psi} = [C_{11} \, C_{12} \, C_{13} \, C_{14} \, C_{21} \, C_{22} \, C_{23} \, C_{24} \, \theta_{k1} \, \theta_{k2}], \tag{28}$$

In order to ensure that the homogeneous Equation (26) has a non-zero solution, the determinant of the characteristic matrix $H(\omega)$ must satisfy:

$$|\boldsymbol{H}(\omega)| = 0, \tag{29}$$

The positive roots obtained by solving Equation (29) are arranged in ascending order, which is the natural frequency of the undamped free vibration of the system: $\omega_1, \omega_2, \omega_3, \ldots \omega_n$ ($r = 1, 2, \ldots, n$). By substituting $\omega_r$ into Equation (26), the corresponding r-order coefficients $C_{ij}$ and $\theta_{ki}$ can be solved, and the obtained coefficients can be substituted into Equation (26) to obtain the r-order modal shape function of the system.

### 2.3. Discrete Dynamic Model of the System Based on Global Mode Method

According to [31], the core of the global modal method is to use only one time coordinate to describe global motion. Specifically, both Equations (13) and (15) contain $sin\omega t$ terms, which means that the motion of rigid bodies and the vibration of flexible bodies are coupled and synchronized.

Therefore, combined with Equations (13) and (15), the displacement of the system is further expressed as:

$$[\theta_s, w_1, w_2, \theta_{k1}, \theta_{k1}]^T = [\theta_{sr}, 0, 0, 0, 0]^T + \boldsymbol{\Phi}\boldsymbol{\eta}(t), \tag{30}$$

Among them, $\boldsymbol{\Phi}$ is the modal matrix, and $\boldsymbol{\eta}(t)$ is the modal coordinate vector. If the first n-order rigid–flexible coupling modes of the system are considered for research, their expressions are as follows:

$$\boldsymbol{\Phi} = \begin{bmatrix} \boldsymbol{\Theta}_{s0} \\ \boldsymbol{\varphi}_1 \\ \boldsymbol{\varphi}_2 \\ \boldsymbol{\Theta}_{k1} \\ \boldsymbol{\Theta}_{k2} \end{bmatrix} = \begin{bmatrix} \theta_{s01}, \theta_{s02}, \dots, \theta_{s0n} \\ \varphi_{11}, \ \varphi_{12}, \dots, \varphi_{1n} \\ \varphi_{21}, \ \varphi_{22}, \dots, \varphi_{2n} \\ \theta_{k11}, \theta_{k12}, \dots, \theta_{k1n} \\ \theta_{k11}, \theta_{k22}, \dots, \theta_{k2n} \end{bmatrix}, \tag{31}$$

$$\boldsymbol{\eta} = [\sin(\omega_1 t), \sin(\omega_2 t), \dots, \sin(\omega_n t)]^T, \tag{32}$$

Substituting Equation (30) into Equation (4) and combining boundary conditions Equations (7) and (8) and hinge matching conditions Equations (11) and (12), the discrete dynamic equation of the system is obtained as follows:

$$\begin{bmatrix} J_t & \boldsymbol{F} \\ \boldsymbol{F}^T & \boldsymbol{M}_\eta \end{bmatrix} \begin{bmatrix} \ddot{\theta}_{sr} \\ \ddot{\boldsymbol{\eta}} \end{bmatrix} + \begin{bmatrix} 0 & 0_{1 \times N} \\ 0_{1 \times N} & \boldsymbol{C}_\eta \end{bmatrix} \begin{bmatrix} \dot{\theta}_{sr} \\ \dot{\boldsymbol{\eta}} \end{bmatrix} + \begin{bmatrix} 0 & 0_{1 \times N} \\ 0_{1 \times N} & \boldsymbol{K}_\eta \end{bmatrix} \begin{bmatrix} \theta_{sr} \\ \boldsymbol{\eta} \end{bmatrix} = \begin{bmatrix} \tau_s \\ \tau_s \boldsymbol{\Theta}_{s0} \end{bmatrix}, \tag{33}$$

where the items of these matrices are expressed as:

$$\boldsymbol{F} = J_t \boldsymbol{\Theta}_{s0} + \rho \left( \int_r^{r+L} x\boldsymbol{\varphi}_1 dx + \int_{-r-L}^{-r} x\boldsymbol{\varphi}_2 dx \right) + J_L(\boldsymbol{\Theta}_{k1} + \boldsymbol{\Theta}_{k2}), \tag{34}$$

$$\begin{aligned}
\boldsymbol{M}_\eta = \quad & J_r \boldsymbol{\Theta}_{s0}^T \boldsymbol{\Theta}_{s0} + \rho \left( \int_r^{r+L} \boldsymbol{\varphi}_1^T \boldsymbol{\varphi}_1 dx + \int_{-r-L}^{-r} \boldsymbol{\varphi}_2^T \boldsymbol{\varphi}_2 dx \right) \\
& + \rho \left[ \int_r^{r+L} x \left( (\boldsymbol{\Theta}_{s0}^T + \boldsymbol{\Theta}_{k1}^T)\boldsymbol{\varphi}_1 + \boldsymbol{\varphi}_1^T (\boldsymbol{\Theta}_{s0} + \boldsymbol{\Theta}_{k1}) \right) dx \right. \\
& \left. + \int_{-r-L}^{-r} x \left( (\boldsymbol{\Theta}_{s0}^T + \boldsymbol{\Theta}_{k2}^T)\boldsymbol{\varphi}_2 + \boldsymbol{\varphi}_2^T (\boldsymbol{\Theta}_{s0} + \boldsymbol{\Theta}_{k2}) \right) dx \right],
\end{aligned} \tag{35}$$

$$\begin{aligned}
\boldsymbol{K}_\eta = \quad & EI_z \int_r^{r+L} [\boldsymbol{\varphi}_1''(x)]^T \boldsymbol{\varphi}_1''(x) dx + EI_z \int_{-r-L}^{-r} [\boldsymbol{\varphi}_2''(x)]^T \boldsymbol{\varphi}_2''(x) dx \\
& + k_L \boldsymbol{\Theta}_{k1}^T \boldsymbol{\Theta}_{k1} + k_L \boldsymbol{\Theta}_{k2}^T \boldsymbol{\Theta}_{k2},
\end{aligned} \tag{36}$$

where $\boldsymbol{C}_\eta = \alpha \boldsymbol{M}_\eta + \beta \boldsymbol{K}_\eta$ is structural damping, and $\alpha$ and $\beta$ are proportional damping coefficients.

## 3. Model and Control Scheme of the Solar Array Drive Assembly

### 3.1. PMSM Model

As mentioned in the introduction, a PMSM direct-drive scheme with a simpler structure and better performance was adopted to provide torque for solar arrays. Assuming that the permanent magnetic flux linkage $\psi_r$ of the rotor in the PMSM is constant, its stator winding voltage can be expressed as [32]:

$$\begin{cases} u_d = (R_s + L_d')i_d - P_n \omega_s L_q i_q \\ u_q = \left( R_s + L_q' \right) i_q + P_n \omega_s L_d i_d + P_n \omega_s \psi_r \end{cases}, \tag{37}$$

where $u_d$ and $u_q$ represent the $d$- and $q$-axis voltages at the stator side, respectively; $i_d$ and $i_q$ represent the $d$- and $q$-axis currents at the stator side, respectively; $R_s$ represents the armature resistance at the stator side; $\omega_s$ indicates the mechanical angular frequency of the rotor; $L_d$ and $L_q$ represent the $d$- and $q$-axis inductances at the stator side, respectively; $L_d'$ and $L_d'$ are the first-order time derivatives of $L_d$ and $L_q$, respectively; $\psi_r$ refers to the magnetic linkage generated by the rotor permanent magnet in the stator winding, that is, the rotor permanent-magnet magnetic linkage; and $p_n$ is the number of rotor poles.

The electromagnetic torque expression of the PMSM is [33]:

$$T_e = \frac{3}{2} P_n \left[ \psi_r i_q + (L_d - L_q) i_d i_q \right], \tag{38}$$

where $T_e$ represents the electromagnetic torque.

In this study, a surface-mounted permanent-magnet synchronous motor (SPMSM) with the same inductance along the d- and q-axes was selected. To ensure the smooth and stable operation of the motor, a field-oriented control (FOC) scheme in which $i_d = 0$ was adopted. In this scheme, the electromagnetic torque formula can be simplified as:

$$T_e = \frac{3}{2} P_n \psi_r i_q. \tag{39}$$

### 3.2. Electromechanical Model of SADA

Figure 3 illustrates the working of the SADA system, which comprises a drive controller, servomotor, and load. The system converts the commands of the central machine into torque and transmits them directly to the solar arrays through the rotary shaft. The acceleration expression of the transmission shaft output is:

$$\ddot{\theta}_m = \frac{1}{J_w} \left( T_e - T_L - T_f \right), \tag{40}$$

where $T_f$ and $T_L$ denote the friction and inertial moments of the load, respectively; $J_m$ is the moment of inertia of the motor; $J_w = J_m + J_t$ represents the moment of inertia of the motor rotor and load without considering flexible vibrations; and $\ddot{\theta}_m$ is the angular acceleration of rotor rotation. Noteworthily, because the motor was directly connected to the rigid shaft without the reducer, the mechanical angle of the rotation of the motor rotor is the same as that of the rigid shaft, that is, $\theta_m = \theta_s$.

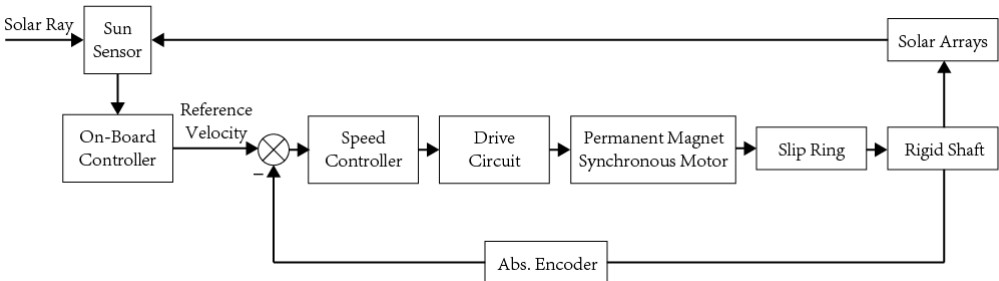

**Figure 3.** Working of a SADA system.

The Stribeck friction model, which is suitable for describing low-speed rotating systems, is typically used in SADA systems. This model is described as:

$$T_f = \sigma_0 \dot{\theta}_m + \left[ T_c + (T_s - T_c) e^{-\left( \frac{\dot{\theta}_m}{\omega_0} \right)^2} \right] sgn\left( \dot{\theta}_m \right), \tag{41}$$

where $\sigma_0$ is the viscous friction coefficient, $T_c$ is the Coulomb friction coefficient, $T_s$ is the maximum static friction moment, and $\omega_0$ is the critical Stribeck speed.

The load torque expression is obtained from Equation (33) as follows:

$$T_L = J_t \ddot{\theta}_m + \boldsymbol{F} \ddot{\boldsymbol{\eta}}, \tag{42}$$

where $\boldsymbol{F}$ and $\boldsymbol{\eta}$ are vectors whose expressions are given by Equation (34) and Equation (32), respectively.

By substituting Equations (39), (41), and (42) into Equation (40), the output equation can be expressed as:

$$\ddot{\theta}_m = \frac{1.5 P_n \psi_r}{J_T} i_q - \frac{T_f}{J_T} - \frac{T_v}{J_T},$$ (43)

where $J_T = J_m + J_t - \boldsymbol{F} \boldsymbol{M}_\eta^{-1} \boldsymbol{F}^T$ represents the total moment of inertia of the motor and load, and $T_v$ represents the impact of the flexible solar array vibration on the drive shaft as follows:

$$T_v = -\boldsymbol{F} \boldsymbol{M}_\eta^{-1} \, \boldsymbol{C}_\eta \dot{\boldsymbol{\eta}} - \boldsymbol{F} \boldsymbol{M}_\eta^{-1} \boldsymbol{K}_\eta \boldsymbol{\eta},$$ (44)

where $T_v$ only exists when a flexible load is driven. When the motor drives a rigid load, the modal coordinate $\boldsymbol{\eta}$ is ignored, and the $T_v$ term in Equation (43) is zero. $i_q$ is the input control current. Both the friction torque and elastic vibration were regarded as system disturbances that could be eliminated by designing control laws.

### 3.3. Design of the DCSMC with the ESO

SMC is essentially a switch control method. In general, the required switching gain needs to be higher than the upper bound of any concentrated disturbance. Therefore, if a disturbance is observed and its impact is compensated by feedforward control, the required switching gain needs to only be higher than the upper limit of the disturbance compensation error, resulting in any system chattering being effectively reduced [34].

Based on this, a DCSMC that could observe disturbances and reduce their impact was adopted to replace the traditional PI speed controller and thereby improve the overall dynamic performance.

Considering the total disturbance effect of the system, the motion equation of the PMSM can be expressed as:

$$\ddot{\theta}_m = \chi i_q^* - f_d,$$ (45)

where $\chi = \frac{3 P_n \psi_r}{2 J_T}$ ; $f_d$ is a function of $\dot{\theta}_m$, which represents the total disturbance of the system that is observed by the ESO and compensated for by feedforward control, resulting in improved control; and $i_q^*$ represents the current signal output by the controller. Owing to the significant difference in bandwidth between the speed and current loops, when adjusting the speed of the outer ring, it is considered that the current in the inner ring has already been adjusted [19]. Therefore, in the design of the speed loop, the effect of the current loop was ignored; that is, $i_q^* = i_q$.

The first-order system state equation can be expressed as:

$$\begin{cases} \dot{x}_1(t) = f(x_1) + h u(t) \\ \quad y(t) = x_1(t) \end{cases},$$ (46)

where $h$ is a constant greater than zero. $f(x_1)$ represents a bounded nonlinear perturbation function, and $u(t)$ is the control input.

If $x_2(t)$ is selected as the expansion variable, and $x_2(t) = f(x_1)$ and $\dot{x}_2(t) = w(t)$, the system expression can be expanded as [35]:

$$\begin{cases} \dot{x}_1(t) = x_2(t) + h u(t) \\ \quad \dot{x}_2(t) = w(t) \\ \quad y(t) = x_1(t) \end{cases},$$ (47)

where if $u(t) = i_q^*$ and $x_1(t) = \dot{\theta}_m$, the ESO based on the hyperbolic tangent function can be described as:

$$\begin{cases} \quad e_1(t) = z_1(t) - \dot{\theta}_m \\ \dot{z}_1(t) = z_2(t) + \chi i_q^* - \beta_1 e_1(t), \\ \quad \dot{z}_2(t) = -\beta_2 \tanh(\beta_3 e_1(t)) \end{cases}$$ (48)

where $z_1(t)$ observes the velocity feedback signal of the PMSM; $z_2(t)$ observes the total disturbance of the system; and $\beta_1$, $\beta_2$, and $\beta_3$ are design parameters that should satisfy $\beta_1 - \beta_2\beta_3 > 0$. The speed-tracking error is defined as:

$$e = \dot{\theta}_m^* - \dot{\theta}_m, \tag{49}$$

where $\dot{\theta}_m^*$ and $\dot{\theta}_m$ represent the given and actual motor speeds, respectively. Combining Equations (49) and (43), the acceleration-tracking error can be expressed as:

$$\dot{e} = \ddot{\theta}_m^* - \chi i_q^* - f_d. \tag{50}$$

In the sliding mode variable-structure control, the design of the controller is typically divided into two parts [36]. The first step is to select the sliding mode surface. The integral sliding surface used in this study is as follows:

$$s = e + c \int_0^t e(\tau)d\tau, \;\; c > 0, \tag{51}$$

The second step involves designing reaching laws. In this study, we designed a VGSRL, which can be expressed as:

$$\dot{s} = -\varepsilon|e|^a sat(s) - k|s|^{b \cdot sgn(|s|-1)}s, \tag{52}$$

where $\varepsilon > 0$, $k > 0$, $0 < a < 1$, $0 < b < 1$. The function $sat(s)$ is defined as [36]:

$$sat(s) = \begin{cases} sgn(s) & |s| > \Delta \\ \frac{s}{\Delta} & |s| < \Delta \end{cases}, \tag{53}$$

where $0 < \Delta \ll 1$ denotes the boundary layer.

The VGSRL is based on the traditional exponential reaching law (TERL), and the system error is introduced as a variable to ensure that the variable gain reaches the sliding surface. We obtained the following conclusions by analyzing the reaching law in Equation (52).

If the system state was far from the sliding surface, that is, if $e$ was large and $s > 1$, then the system state approached the sliding surface at a rate of $\varepsilon|e|^a$ and $k|s|^b s$. When the system state gradually reached the sliding mode surface, the errors $e$ and $s$ gradually decreased, and the variable-gain reaching speed $\varepsilon|e|^a$ and variable-index reaching speed $k|s|^b s$ also decreased. In other words, as the system state approached the sliding surface, the reaching speed automatically decreased to reduce chattering. When the system state entered the boundary layer ($|s| < \Delta$), linear feedback control was used instead of $sgn(s)$ to reduce chattering. However, when $s < 1$, the reaching speed of the TERL decreased to zero, resulting in the speed of the system decreasing to reach a steady state. Since $sgn(|s| - 1)$ was adopted in the index, the variable-index reaching speed was $k|s|^{-b}s$, which was larger than $k|s|^b s$ under the TERL and could reach the sliding surface faster.

In summary, by combining Equations (50)–(52), the signal output of the speed controller, namely, the value of the reference current of the q-axis, can be expressed as:

$$i_q^* = \frac{1}{\chi}\left[\ddot{\theta}_m^* - f_d + \varepsilon|e|^a sgn(s) + k|s|^{b \cdot sgn(|s|-1)}s + ce\right], \tag{54}$$

where $-f_d/\chi$ represents the impact compensated for by feedforward control based on the ESO.

*3.4. Stability Proof*

The Lyapunov function can be constructed as:

$$V = \frac{1}{2}s^2, \tag{55}$$

By combining Equations (50)–(52), the derivation of Equation (53) is obtained as follows:

$$
\begin{aligned}
\dot{V} = s\dot{s} &= s\left(\dot{e} + ce\right) \\
&= s\left(\ddot{\theta}_m^* - \chi i_q^* - f_d + ce\right) \\
&= -s\left[\varepsilon|e|^a sat(s) + k|s|^{b \cdot sgn(|s|-1)}s\right] \leq 0,
\end{aligned}
\tag{56}
$$

When the parameter selection satisfies $\varepsilon > 0$, $k > 0$, $0 < a < 1$, and $0 < b < 1$, then $\dot{V} \leq 0$ is obtained. Then, according to the Lyapunov stability theory, the system is asymptotically stable.

## 4. Numerical Results and Discussion

*4.1. Validation of the Dynamic Model*

The geometric parameters and material properties of the motor shaft with the pair of solar arrays are listed in Table 1. To verify the validity and accuracy of the model obtained by the GMM, the results were compared with those obtained by using the commercial finite element software ANSYS. Figure 4 shows the finite element model of the SSH in ANSYS. The rigid-axis model was built with a body element, the flexible solar array was made of a shell element, and the flexible hinge connecting them was expressed using multipoint constraints (MPCs).

**Table 1.** The structural parameters of the SADA system.

| Components | Parameters | Values |
|---|---|---|
| Solar arrays | Length $L$ (m) | 4.0 |
| | Width $h$ (m) | 0.3 |
| | Thickness of solar array $b$ (m) | 0.01 |
| | Elastic modulus of aluminum | $7 \times 10^{10}$ |
| | Mass density of aluminum $\rho_0$ (kg m$^{-3}$) | 2700 |
| | Poisson ratio $\mu$ | 0.33 |
| Hinge | Torsional rigidity $k_L$ (Nm/rad) | 400 |
| Rigid shaft | Radius $r$ (m) | 0.01 |

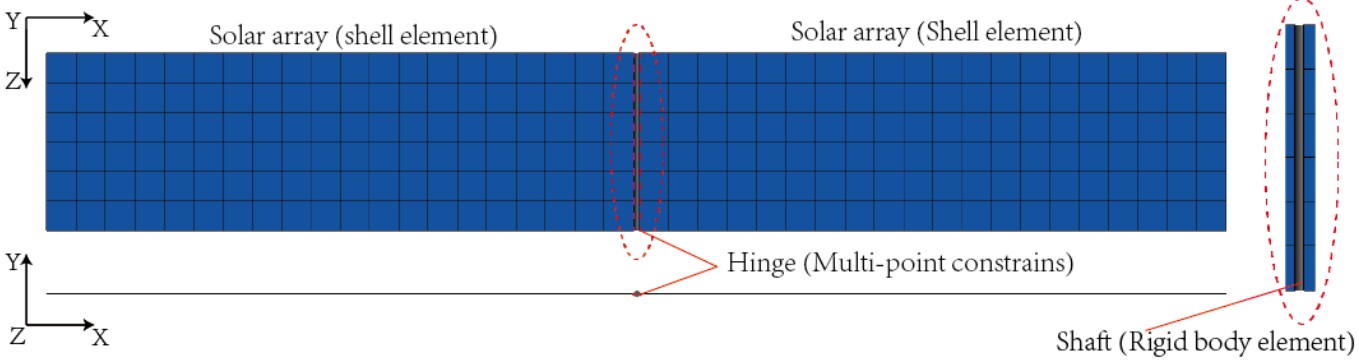

**Figure 4.** Finite element model of the shaft and solar arrays with hinge.

Figure 5 shows the first six bending mode shapes of the SSH structure solved by using the GMM and ANSYS when $L = 2\ m$. The modal shapes obtained by using the two

methods were clearly consistent. The first-, third-, and fifth-order modes were positively symmetric, and the second-, fourth-, and sixth-order modes were antisymmetric.

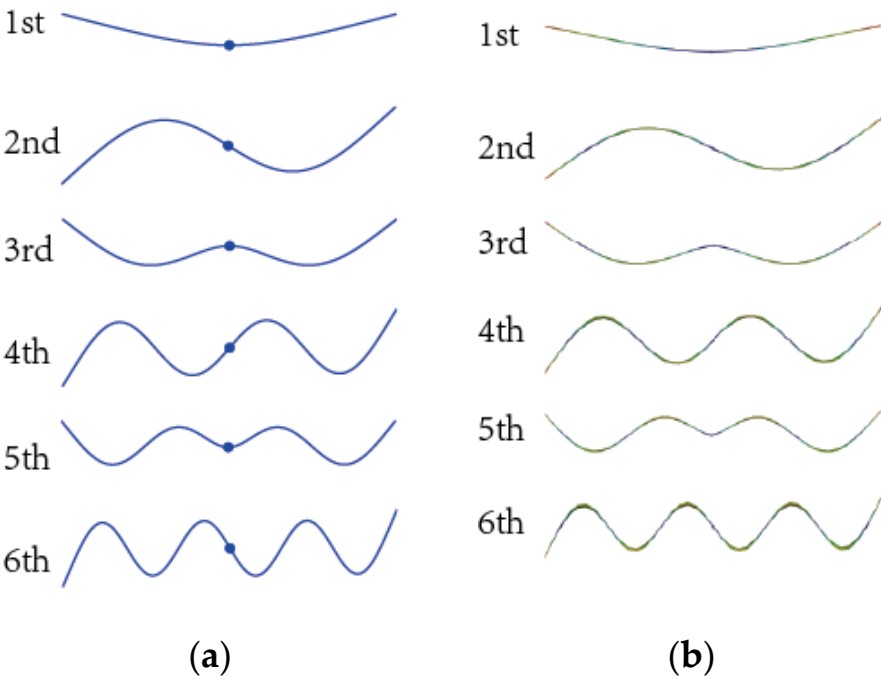

**Figure 5.** The first six orders of bending modes of the solar array; (**a**) the modes with GMM; (**b**) the modes with FEM.

Tables 2–4 show the first frequencies of the SSH model at different $L$, $J_s$, and $k_L$ values. The relative error (*Re*) between the analytical solution obtained by using the GMM and the numerical solution obtained by using ANSYS are also listed. *Re* can be expressed as:

$$Re = \frac{f_{cal} - f_{FEM}}{f_{FEM}}, \tag{57}$$

where $f_{cal}$ and $f_{FEM}$ are the results calculated using Equation (29) and ANSYS, respectively. According to Tables 2–4, the maximum absolute value of *Re* is 1.93%, which indicates the high accuracy of the model with the GMM.

**Table 2.** Frequencies of the first six bending modes with different $L$ (Hz).

| Modal Order * | L=2 m | | | L=4 m | | | L=8 m | | |
|---|---|---|---|---|---|---|---|---|---|
| | Ansys | GMM | Re (%) | Ansys | GMM | Re (%) | Ansys | GMM | Re (%) |
| 1 | 1.941 | 1.971 | −1.55% | 0.504 | 0.509 | −0.99% | 0.124 | 0.126 | −1.61% |
| 2 | 9.038 | 9.029 | 0.10% | 2.25 | 2.233 | 0.76% | 0.556 | 0.557 | −0.18% |
| 3 | 12.479 | 12.72 | −1.93% | 3.169 | 3.192 | −0.73% | 0.778 | 0.788 | −1.29% |
| 4 | 29.82 | 29.448 | 1.25% | 7.369 | 7.254 | 1.56% | 1.822 | 1.805 | 0.93% |
| 5 | 35.524 | 34.978 | 1.54% | 8.965 | 8.924 | 0.46% | 2.129 | 2.104 | 1.17% |
| 6 | 63.212 | 62.774 | 0.69% | 15.629 | 15.34 | 1.85% | 3.829 | 3.776 | 1.38% |

* ($J_s = 0.078 \, kgm^2 \, k_L = 400 \, Nm/rad$).

**Table 3.** Frequencies of the first six bending modes with different $J_s$ (Hz).

| Modal Order * | $J_s$=0.078 $kgm^2$ | | | $J_s$=0.78 $kgm^2$ | | | $J_s$=7.8 $kgm^2$ | | |
|---|---|---|---|---|---|---|---|---|---|
| | Ansys | GMM | Re (%) | Ansys | GMM | Re (%) | Ansys | GMM | Re (%) |
| 1 | 0.124 | 0.126 | −1.61% | 0.124 | 0.126 | −1.61% | 0.124 | 0.126 | −1.61% |
| 2 | 0.556 | 0.557 | −0.18% | 0.556 | 0.557 | −0.18% | 0.556 | 0.557 | −0.18% |
| 3 | 0.778 | 0.788 | −1.29% | 0.778 | 0.788 | −1.29% | 0.778 | 0.788 | −1.29% |
| 4 | 1.822 | 1.805 | 0.93% | 1.822 | 1.805 | 0.93% | 1.822 | 1.805 | 0.93% |
| 5 | 2.129 | 2.104 | 1.17% | 2.129 | 2.104 | 1.17% | 2.129 | 2.104 | 1.17% |
| 6 | 3.829 | 3.776 | 1.38% | 3.829 | 3.776 | 1.38% | 3.829 | 3.776 | 1.38% |

* ($L = 2\ m, k_L = 600\ Nm/rad$).

**Table 4.** Frequencies of the first six bending modes with different $k_L$ (Hz).

| Modal Order * | $k_L$=300 $Nm/rad$ | | | $k_L$=800 $Nm/rad$ | | | $k_L$=1200 $Nm/rad$ | | |
|---|---|---|---|---|---|---|---|---|---|
| | Ansys | GMM | Re (%) | Ansys | GMM | Re (%) | Ansys | GMM | Re (%) |
| 1 | 1.868 | 1.859 | 0.48% | 2.058 | 2.071 | −0.63% | 2.081 | 2.101 | −0.96% |
| 2 | 9.038 | 9.029 | 0.10% | 9.038 | 9.029 | 0.10% | 9.038 | 9.029 | 0.10% |
| 3 | 11.889 | 12.014 | −1.05% | 12.935 | 13.026 | −0.70% | 13.076 | 13.165 | −0.68% |
| 4 | 29.82 | 29.448 | 1.25% | 29.82 | 29.448 | 1.25% | 29.82 | 29.448 | 1.25% |
| 5 | 33.993 | 33.773 | 0.65% | 36.741 | 36.174 | 1.54% | 37.149 | 36.864 | 0.77% |
| 6 | 63.212 | 62.774 | 0.69% | 63.212 | 62.774 | 0.69% | 63.212 | 62.774 | 0.69% |

* ($L = 2\ m, J_s = 0.078\ kgm^2$).

The natural frequency decreased with an increase in the length of the solar array, indicating the increased density and flexibility of the system mode. When the moment of inertia of the rigid shaft $J_s$ increased, the frequencies of orders 1, 3, and 5 remained unchanged, and the frequencies of orders 2, 4, and 6 decreased, indicating the coupling of the antisymmetric mode with the rigid shaft and the lack of coupling between the positive symmetric mode and the rigid shaft. In contrast, when the torsional stiffness of the hinge increased, the modal frequencies of orders 1, 3, and 5 increased, and the modal frequencies of orders 2, 4, and 6 remained almost unchanged, indicating that the stiffness of the flexible joint had affected the flexibility of the system but hardly affected the coupling between the rigid and flexible bodies.

### 4.2. Numerical Simulation Results of the Driving Process

Figure 6 shows the disturbance torque response curves for different windsurfer sizes from zero acceleration to 0.06 °/s. The remaining attribute parameters of the solar array are listed in Table 1. In general, when there were no increases in the external disturbances, the disturbance torque gradually decayed to zero. Figure 6a shows that as the length of the sail increased, the disturbance frequency decreased, the amplitude increased, and the attenuation rate decreased. Figure 6b shows that as the thickness of the sail increased, the frequency of the disturbance and attenuation speed increased, indicating an increase in modal damping. However, the amplitude of the disturbance also increased simultaneously owing to the larger thickness, resulting in a greater moment of inertia.

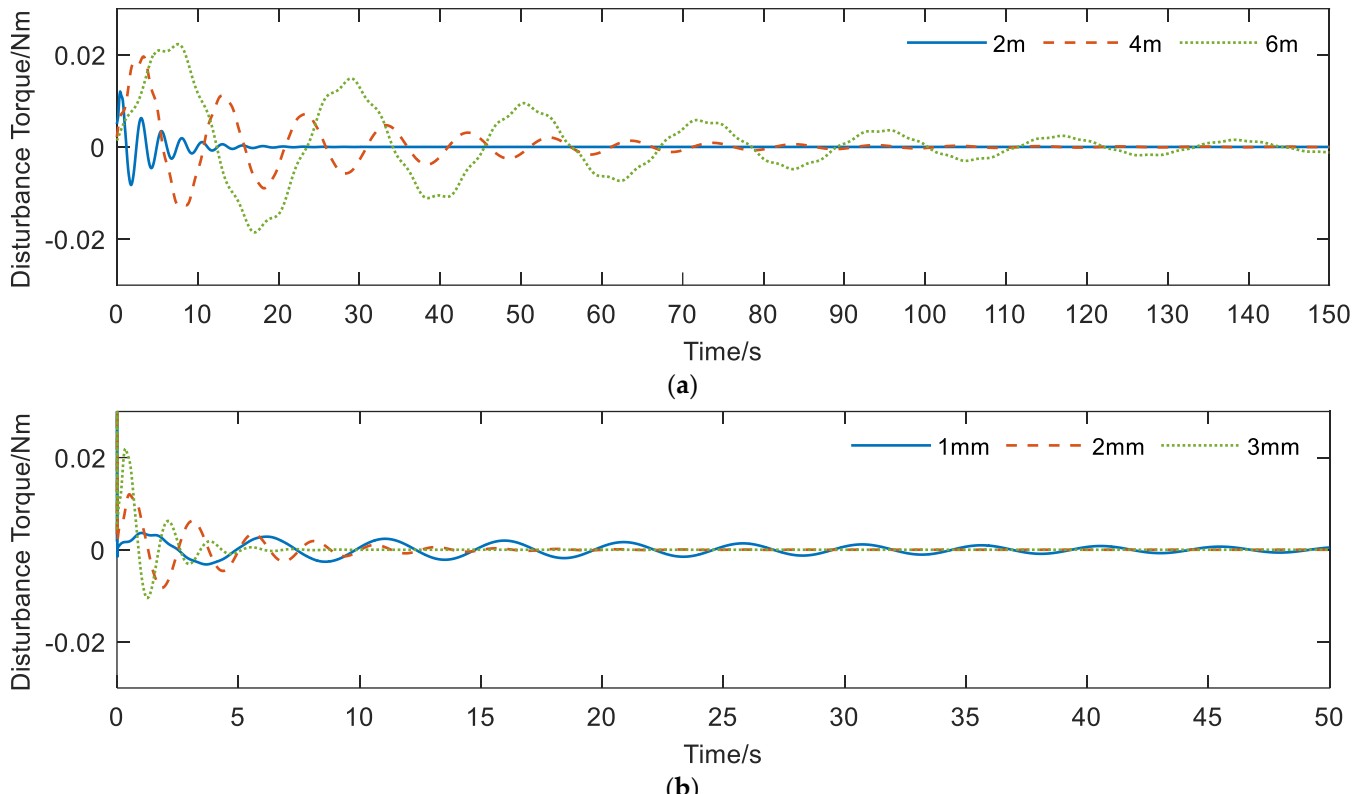

**Figure 6.** (**a**) Disturbance torque response of different *L*; (**b**) disturbance torque response of different *b*.

Figure 7 shows the disturbance torque curve considering the SVPWM current harmonics. The amplitude of the disturbance torque oscillation gradually decayed and ultimately stabilized at a lower level. The coupling of the flexible torque and current harmonics increased the amplitude of the disturbance torque and made attenuation difficult. Therefore, it was necessary to design a control law to improve these results.

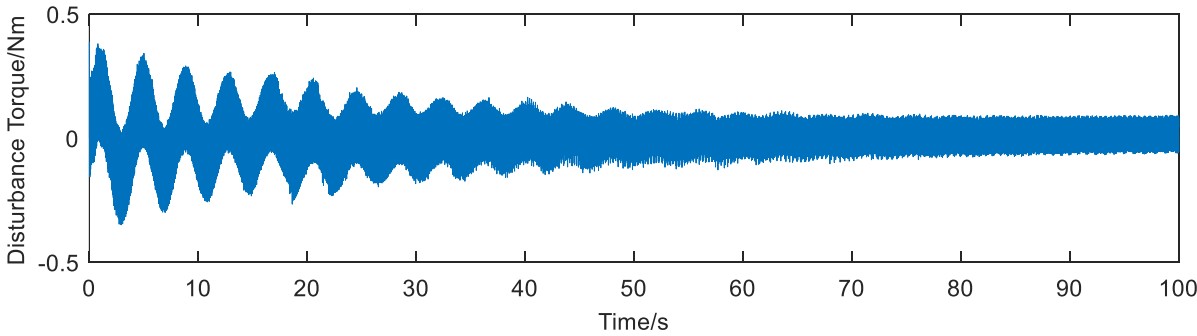

**Figure 7.** Disturbance torque with current harmonics.

This section presents the results of the simulation and comparison with PI and SMC strategies that were performed to verify the superiority of the control performance of the proposed algorithm. PI control adopted a two-closed-loop structure that included speed and current loops [37], while SMC adopted the traditional exponential reaching law shown in Equation (58) and the integral sliding surface shown in Equation (59) [38].

$$\dot{s} = -d_s sgn(s) - k_s s, \tag{58}$$

$$s = e + c_s \int_0^t e(\tau) d\tau, \quad c > 0, \tag{59}$$

The parameters of PI control were designed via the root locus method, while the parameters of SMC and DCSMC were iteratively tuned using a trial-and-error method based on the system performance, and the parameter selection method for the ESO was based on the method in reference [35]. Equation (52) shows that when $a = 0$, $b = 0$, the VGSRL will collapse into the traditional exponential reaching law, and the parameters $c, k,$ and $\varepsilon$ in DCSMC will correspond to parameters $c_s$, $k_s,$ and $d_s$ in SMC. It was easy to note that the large parameters could lead to a large gain, which could accelerate the vibration suppression. However, if the parameters were too large, a divergence in the computation may result. The specific values could be calibrated by simulation results; thus, we iteratively tuned the parameters $a$ and $b$ by increasing their value from zero and tuned $c, k,$ and $\varepsilon$ by decreasing their values from a small positive number. The parameters of SMC were also determined through a trial-and-error method. In this work, the parameters of the three controllers were set as in Table 5. The parametric inaccuracy, external disturbance torque, and all control parameters of the current loop in the three cases remained the same.

**Table 5.** The parameters of three controllers.

| Controller | Parameters | Values |
|---|---|---|
| PI control | Proportional coefficient (Speed loop) | 2 |
| | Integral coefficient (Speed loop) | 20 |
| | Proportional coefficient (Current loop) | 20 |
| | Integral coefficient (Current loop) | 20 |
| SMC | $c_s$ | 2 |
| | $k_s$ | 2.5 |
| | $d_s$ | 2.8 |
| DCSMC | $a$ | 0.45 |
| | $b$ | 0.65 |
| | $c$ | 20 |
| | $\varepsilon$ | 5 |
| | $k$ | 23 |
| | $\beta_1$ | 160 |
| | $\beta_2$ | 160 |
| | $\beta_3$ | 0.94 |

Figure 6 clearly shows that the characteristics of disturbance torque depend on the physical parameters of the solar array. To make a clear comparison, the physical parameters of the solar array and the motor were selected, as shown in Tables 1, 5 and 6. When the satellite operated in the solar-synchronous orbit, the solar arrays needed to rotate at a constant speed of 0.06 $°/s$ to be oriented toward the Sun. Thus, the driving speed of this simulation was selected as 0.06 $°/s$.

**Table 6.** The parameters of the PMSM.

| Parameters | Values |
|---|---|
| Pole pairs $P_n$ | 32 |
| Flux linkage $\psi_r$ (*Wb*) | 0.0625 |
| Inductance of $q$-axis $L_q$ (*mH*) | 5 |
| Inductance of $d$-axis $L_d$ (*mH*) | 5 |
| Armature resistance $R_s$ (*Ohm*) | 2.25 |
| Bus voltage $V_{dc}$ (*V*) | 28 |
| Rotor inertia $J_m$ (*kg m²*) | 0.01 |
| Maximum output torque $T_{max}$ (*Nm*) | 4 |

Figure 8 shows the rotation angle, angular velocity, disturbance torque, and solar tip elastic displacement curves of the SADA driven by the PI, SMC, and DCSMC strategies. Given the initial signal at 0 *s*, the system accelerated from zero to 0.06 °/*s* at a constant speed.

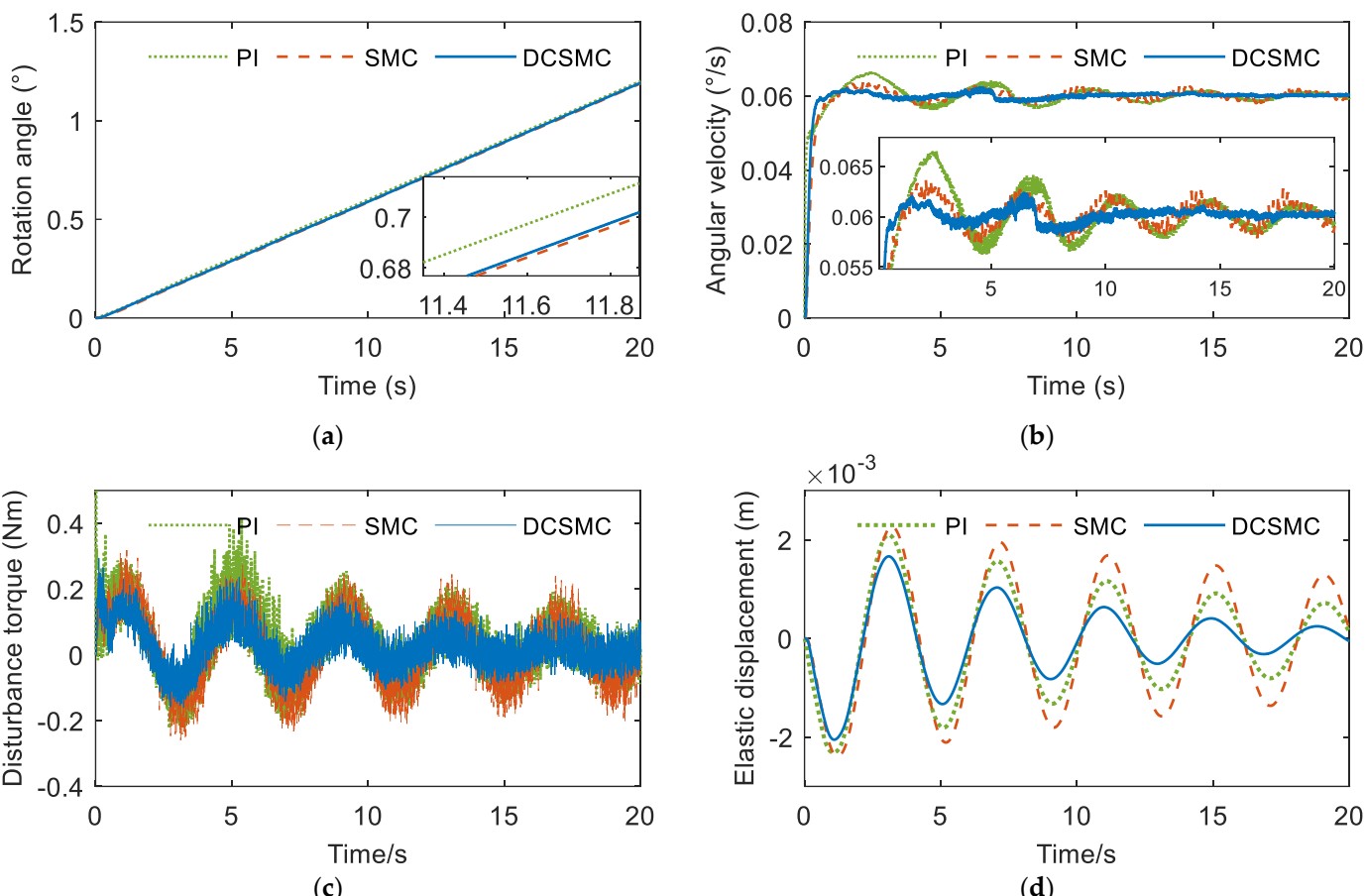

**Figure 8.** Drive response curve of constant speed: (**a**) rotation angle; (**b**) angular velocity; (**c**) disturbance torque; (**d**) solar tip elastic displacement.

In Figure 8a, the three curves almost completely coincide, indicating that all three algorithms could effectively control the solar array such that it would reach the designated position. Figure 8b shows that the PI method had a larger speed overshoot than the SMC method did; however, its oscillation decayed faster owing to the inherent high-frequency chattering effect of the SMC method. The DCSMC method adopted a reaching law to suppress chattering and compensate for the disturbance torque, resulting in a lower speed overshoot and faster oscillation attenuation compared to those using the other two methods. Figure 8c,d show that using the DCSMC, the amplitude of disturbance torque was smaller, and the decay rate was faster than those with the other two methods.

In some scenarios, solar arrays need to quickly maneuver to a certain position or the operating speed needs to be changed, and any change in speed will cause changes in torque and affect the operation of the overall satellite. Considering these conditions, the speed regulation process of SADA was simulated. Figure 9 shows the motor starting, speed switching, and stopping curves. The speed command changed from 0.06 to 0.3 °/*s* at the 10th second, and the motor stopped working at the 20th second.

Figure 9b clearly shows that the system using the PI and SMC methods exhibited a significant speed overshoot and oscillation after speed regulation, while the DCSMC method had a smaller overshoot and faster oscillation attenuation. When the speed com-

mand changed from 0.06 to 0.3 °/*s*, the overshoots of the PI, SMC, and DCSMC schemes were 0.027, 0.015, and 0.006 °/*s*, respectively. The DCSMC method reduced the overshoot by 77.8 and 60%, respectively, compared to the reductions obtained by the PI and SMC methods. Figure 9c shows that the PI method, owing to its large gain, triggered a large instantaneous disturbance torque, which was detrimental to the stable operation of the system. The disturbance torque excited by the DCSMC was relatively small and exhibited the fastest decay rate; the same conclusion was drawn from Figure 9d.

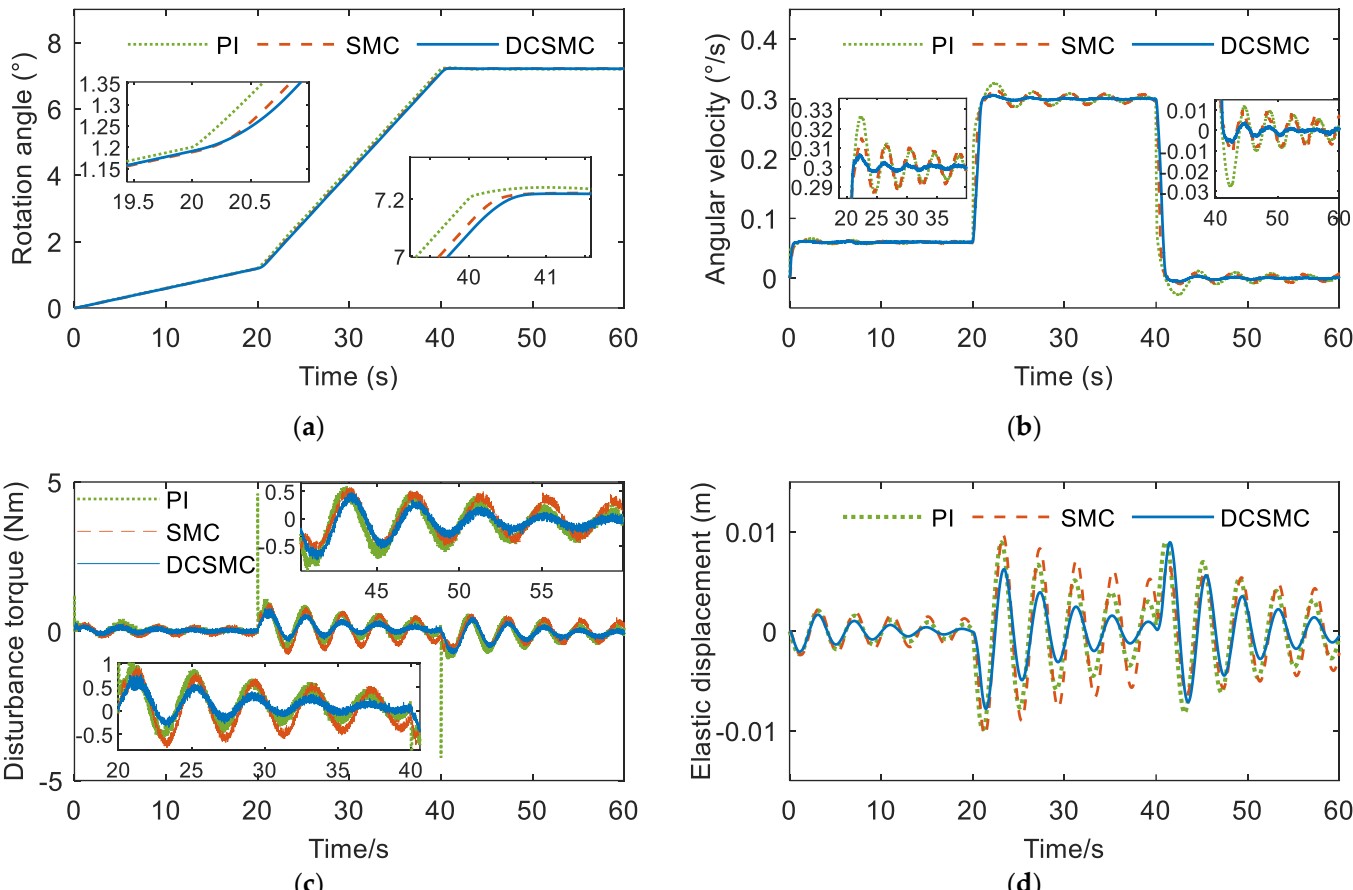

**Figure 9.** Drive response curve of speed regulation: (**a**) rotation angle; (**b**) angular velocity; (**c**) disturbance torque; (**d**) solar tip elastic displacement.

Satellites are affected by various disturbances when operating in orbit; thus, simulating the operation of the system under external disturbances is necessary. Figure 10 shows the curve of the system subjected to external disturbances during operation. The speed command was set to 0.06 °/*s* at 0 *s*, and the step additional disturbance torque was increased to $\tau_d = 0.5$ *Nm* at the 10th second to test the dynamic performance of the system.

As shown in Figure 9b, the absolute values of velocity fluctuations after external disturbances in the DCSMC, SMC, and PI schemes were 0.007, 0.018, and 0.013°/*s*, respectively. The DCSMC method reduced the overshoot by 46.1 and 61.1%, respectively, compared to the reductions obtained by the PI and SMC methods, which indicates better speed stability and robustness.

During the actual operation of the system, motor or structural parameters can be changed; thus, comparing the robustness of the control methods is necessary. This study used the Gaussian random torque to simulate disturbances caused by parameter changes. Figure 11 shows the random disturbance torque.

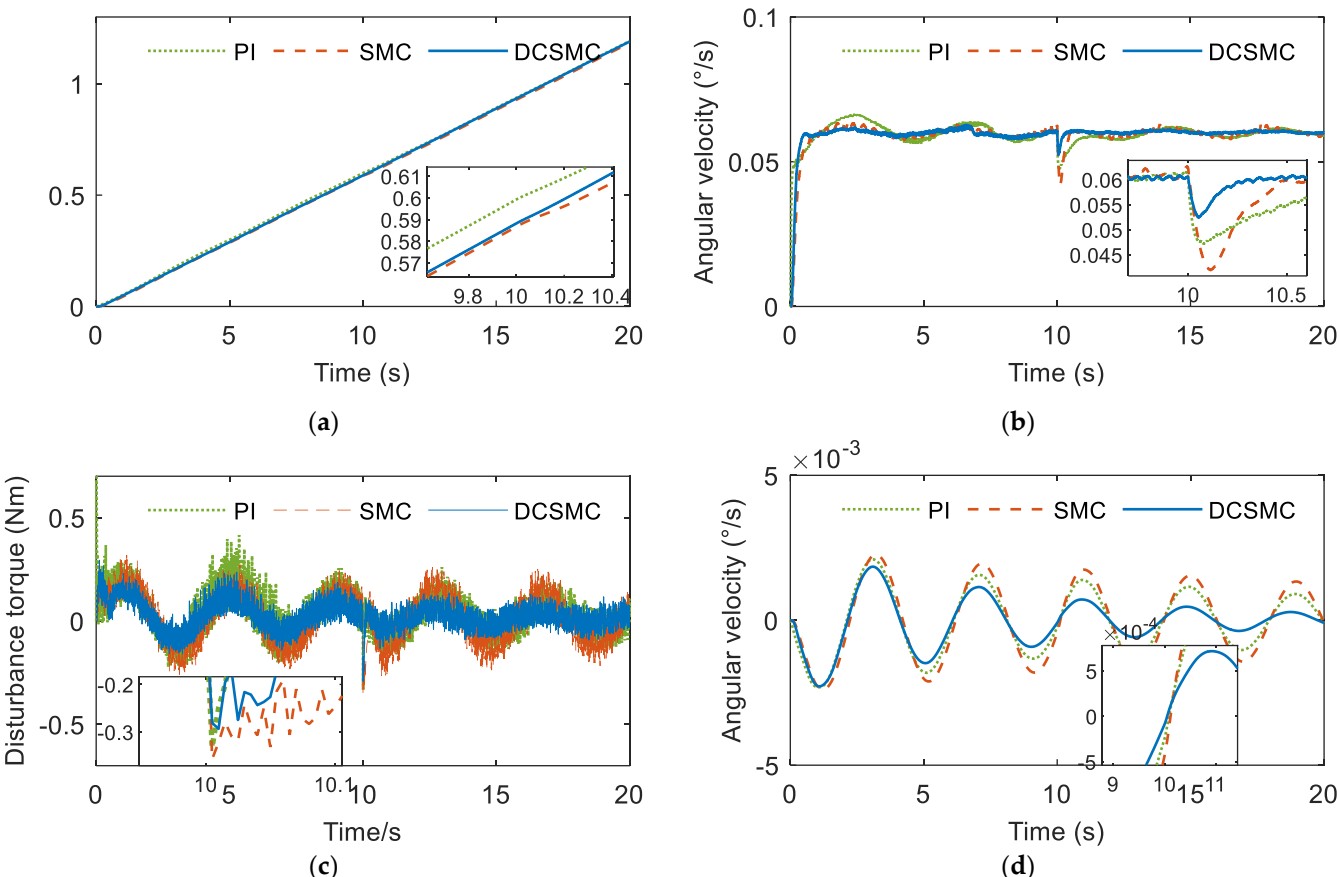

**Figure 10.** Drive response curve under step disturbance: (**a**) rotation angle; (**b**) angular velocity; (**c**) disturbance torque; (**d**) solar tip elastic displacement.

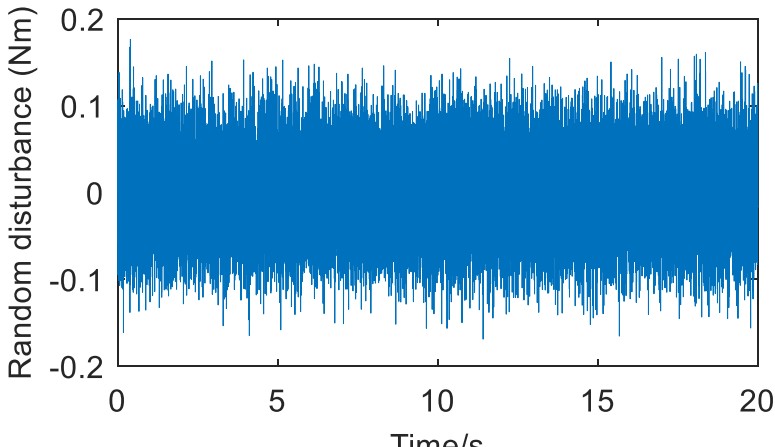

**Figure 11.** Random disturbance torque waveform.

Figure 12b clearly shows that the system using the PI and SMC methods exhibited significant speed chatter under the influence of random disturbances. In Figure 12c, the high-frequency disturbance torque of PI and SMC significantly increased, while the torque when using DCSMC had almost no change, once again demonstrating the high robustness of this method. In Figure 12d, the elastic displacement of the solar array under the control of three methods is shown to increase to varying degrees, but the system under the control of the DCSMC method had the smallest elastic displacement and the fastest decay speed. This indicates that DCSMC has better robustness under the influence of unknown high-frequency disturbances.

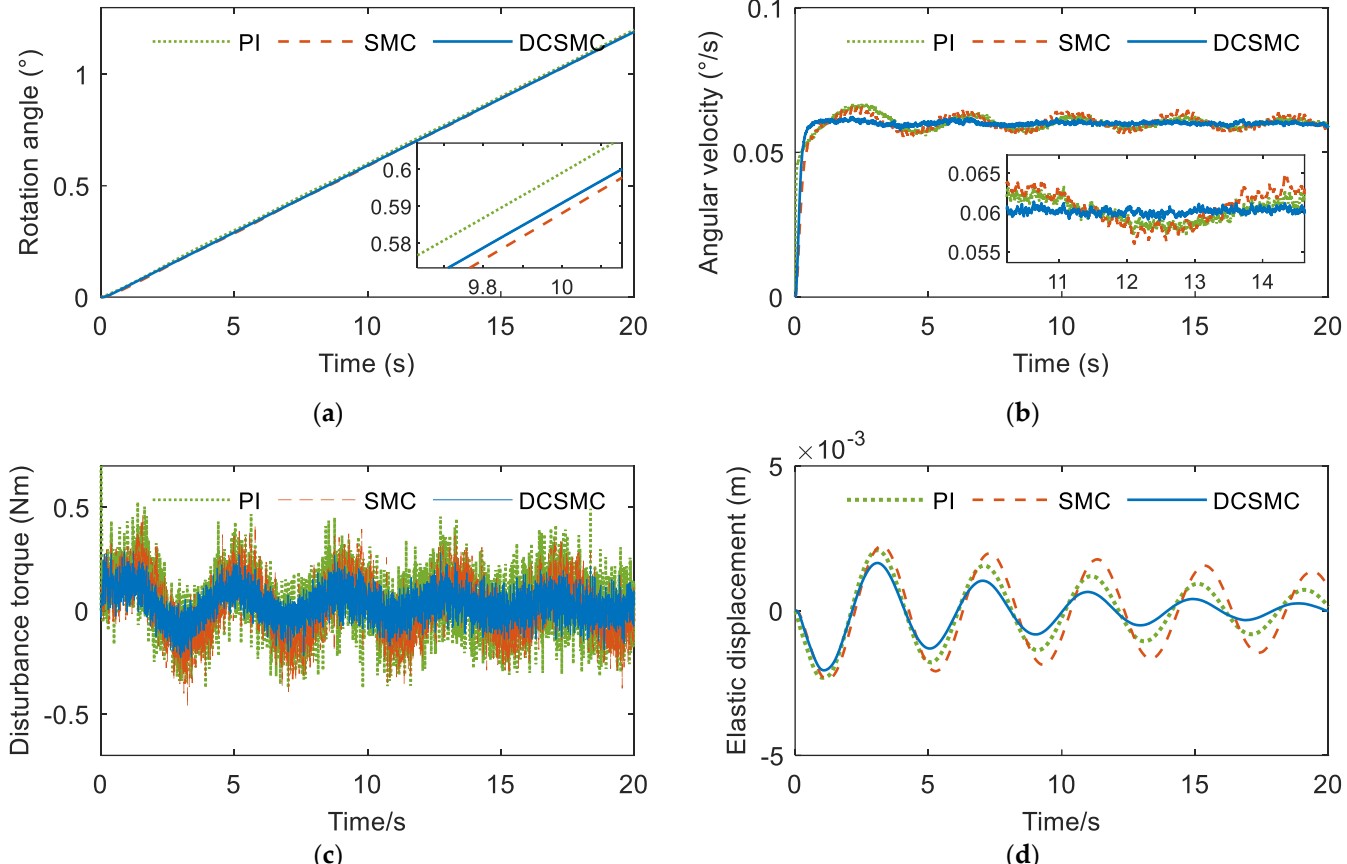

**Figure 12.** Drive response curve under random disturbance: (**a**) rotation angle; (**b**) angular velocity; (**c**) disturbance torque; (**d**) solar tip elastic displacement.

## 5. Conclusions

In this study, a SADA model was established, and a high-stability control strategy was proposed to suppress vibration. A linearization model of a flexible array with a rigid-shaft flexible connection was established and simulated. The comparison of the simulation results with those of the commercial finite element software ANSYS verified the effectiveness of the global model obtained by the GMM.

A variable-gain saturation reaching law was proposed, and a disturbance compensation SMC strategy was designed based on the ESO. The stability of the system was proved using the Lyapunov theory. Low-speed simulations showed that the system controlled by the DCSMC method had a better performance in the settling time, the overshoot, and the tracking error of angular velocity control. On the basis of this paper, compared with the PI and SMC methods, DCSMC can reduce the overshoot of angular velocity by 77.8 and 60%, respectively. Additionally, the system controlled by the DCSMC method had a lower disturbance torque amplitude and faster disturbance torque attenuation under uniform step and random disturbance conditions.

Therefore, disturbance compensation sliding mode control can suppress the flexible vibration and improve the angular velocity control performance.

**Author Contributions:** Conceptualization, J.L. and L.G.; methodology, J.L.; software, J.L.; validation, J.L., H.J. and L.-B.K.; formal analysis, H.H.; investigation, J.L.; resources, M.-S.C.; data curation, M.-S.C.; writing—original draft preparation, J.L.; writing—review and editing, M.-S.C.; visualization, L.-B.K.; supervision, L.-B.K.; project administration, L.G.; funding acquisition, L.G. All authors have read and agreed to the published version of the manuscript.

**Funding:** This research received no external funding.

**Data Availability Statement:** Not applicable.

**Conflicts of Interest:** The authors declare no conflict of interest.

**Appendix A**

$$\xi_{11}(\lambda) = \int_r^{r+L} \rho x \cosh(\lambda x)dx, \ \xi_{12}(\lambda) = \int_r^{r+L} \rho x \sinh(\lambda x)dx,$$

$$\xi_{13}(\lambda) = \int_r^{r+L} \rho x \cos(\lambda x)dx, \ \xi_{14}(\lambda) = \int_r^{r+L} \rho x \sin(\lambda x)dx,$$

$$\xi_{21}(\lambda) = \int_{-r-L}^{-r} \rho x \cosh(\lambda x)dx, \ \xi_{22}(\lambda) = \int_{-r-L}^{-r} \rho x \sinh(\lambda x)dx,$$

$$\xi_{23}(\lambda) = \int_{-r-L}^{-r} \rho x \cos(\lambda x)dx, \ \xi_{24}(\lambda) = \int_{-r-L}^{-r} \rho x \sin(\lambda x)dx,$$

$$B_{11}(x) = \cosh(\lambda x) + \frac{x}{J_t}\xi_{11}(\lambda), \ B_{12}(x) = \sinh(\lambda x) + \frac{x}{J_t}\xi_{12}(\lambda),$$

$$B_{13}(x) = \cos(\lambda x) + \frac{x}{J_t}\xi_{13}(\lambda), \ B_{14}(x) = \sin(\lambda x) + \frac{x}{J_t}\xi_{14}(\lambda),$$

$$B_{21}(x) = \cosh(\lambda x) + \frac{x}{J_t}\xi_{21}(\lambda), \ B_{22}(x) = \sinh(\lambda x) + \frac{x}{J_t}\xi_{22}(\lambda),$$

$$B_{23}(x) = \cos(\lambda x) + \frac{x}{J_t}\xi_{23}(\lambda), \ B_{24}(x) = \sin(\lambda x) + \frac{x}{J_t}\xi_{24}(\lambda),$$

$$b_1(x) = C_{21}\xi_{21}(\lambda)\frac{x}{J_t} + C_{22}\xi_{22}(\lambda)\frac{x}{J_t} + C_{23}\xi_{23}(\lambda)\frac{x}{J_t} + C_{24}\xi_{24}(\lambda)\frac{x}{J_t} + \frac{J_L x}{J_t}\theta_{k1},$$

$$b_2(x) = C_{11}\xi_{11}(\lambda)\frac{x}{J_t} + C_{12}\xi_{12}(\lambda)\frac{x}{J_t} + C_{13}\xi_{13}(\lambda)\frac{x}{J_t} + C_{14}\xi_{14}(\lambda)\frac{x}{J_t} + \frac{J_L x}{J_t}\theta_{k2},$$

$$H(\omega) = \begin{bmatrix} H_{0101} & H_{0102} & H_{0103} & H_{0104} & H_{0105} & H_{0106} & H_{0107} & H_{0108} & H_{0109} & H_{0110} \\ H_{0201} & H_{0202} & H_{0203} & H_{0204} & H_{0205} & H_{0206} & H_{0207} & H_{0208} & H_{0209} & H_{0210} \\ H_{0301} & H_{0302} & H_{0303} & H_{0304} & 0 & 0 & 0 & 0 & H_{0309} & 0 \\ H_{0401} & H_{0402} & H_{0403} & H_{0404} & 0 & 0 & 0 & 0 & 0 & 0 \\ H_{0501} & H_{0502} & H_{0503} & H_{0504} & 0 & 0 & 0 & 0 & 0 & 0 \\ H_{0601} & H_{0602} & H_{0603} & H_{0604} & H_{0605} & H_{0606} & H_{0607} & H_{0608} & H_{0609} & H_{0610} \\ H_{0701} & H_{0702} & H_{0703} & H_{0704} & H_{0705} & H_{0706} & H_{0707} & H_{0708} & H_{0709} & H_{0710} \\ 0 & 0 & 0 & 0 & H_{0806} & H_{0807} & H_{0808} & H_{0809} & 0 & H_{0810} \\ 0 & 0 & 0 & 0 & H_{0906} & H_{0907} & H_{0908} & H_{0909} & 0 & 0 \\ 0 & 0 & 0 & 0 & H_{1006} & H_{1007} & H_{1008} & H_{1009} & 0 & 0 \end{bmatrix},$$

$$H_{0101} = \cosh(\lambda r) + \frac{r}{J_t}\xi_{11}(\lambda), H_{0102} = \sinh(\lambda r) + \frac{r}{J_t}\xi_{12}(\lambda),$$

$$H_{0103} = \cos(\lambda r) + \frac{r}{J_t}\xi_{13}(\lambda), H_{0104} = \sin(\lambda r) + \frac{r}{J_t}\xi_{14}(\lambda),$$

$$H_{0105} = \frac{r}{J_t}\xi_{21}(\lambda), H_{0106} = \frac{r}{J_t}\xi_{22}(\lambda),$$

$$H_{0107} = \frac{r}{J_t}\xi_{23}(\lambda), H_{0108} = \frac{r}{J_t}\xi_{24}(\lambda),$$

$$H_{0109} = \frac{J_L r}{J_t}, H_{0110} = \frac{J_L r}{J_t},$$

$$H_{0201} = \lambda \sinh(\lambda r) + \frac{\xi_{11}(\lambda)}{J_t}, H_{0202} = \lambda \cosh(\lambda r) + \frac{\xi_{12}(\lambda)}{J_t},$$

$$H_{0203} = -\lambda \sin(\lambda r) + \frac{\xi_{13}(\lambda)}{J_t}, H_{0204} = \lambda \cos(\lambda r) + \frac{\xi_{14}(\lambda)}{J_t},$$

$$H_{0205} = \frac{\xi_{21}(\lambda)}{J_t}, H_{0206} = \frac{\xi_{22}(\lambda)}{J_t},$$

$$H_{0207} = \frac{\xi_{23}(\lambda)}{J_t}, \ H_{0208} = \frac{\xi_{24}(\lambda)}{J_t},$$

$$H_{0209} = \frac{J_L - J_t}{J_t}, H_{0210} = \frac{J_L - J_t}{J_t},$$

$$H_{0301} = \lambda^2 \cosh(\lambda r), H_{0302} = \lambda^2 \sinh(\lambda r),$$

$$H_{0303} = -\lambda^2 \cos(\lambda r), H_{0304} = -\lambda^2 \sin(\lambda r), \ H_{0309} = -\frac{K_L}{EI_Z},$$

$$H_{0401} = \lambda^2 \cosh(\lambda(r + L)), H_{0402} = \lambda^2 \sinh(\lambda(r + L)),$$

$$H_{0403} = -\lambda^2 \cos(\lambda(r+L)), H_{0404} = -\lambda^2 \sin(\lambda(r+L)),$$

$$H_{0501} = \lambda^3 \sinh(\lambda(r+L)), H_{0502} = \lambda^3 \cosh(\lambda(r+L)),$$

$$H_{0503} = \lambda^3 \sinh(\lambda(r+L)), H_{0504} = \lambda^3 \cosh(\lambda(r+L)),$$

$$H_{0601} = \frac{-r}{J_t}\xi_{11}(\lambda), H_{0602} = \frac{-r}{J_t}\xi_{12}(\lambda),$$

$$H_{0603} = \frac{-r}{J_t}\xi_{13}(\lambda), H_{0108} = \frac{-r}{J_t}\xi_{14}(\lambda),$$

$$H_{0605} = \cosh(-\lambda r) + \frac{-r}{J_t}\xi_{21}(\lambda), H_{0606} = \sinh(\lambda r) + \frac{-r}{J_t}\xi_{22}(\lambda),$$

$$H_{0607} = \cos(\lambda r) + \frac{-r}{J_t}\xi_{23}(\lambda), H_{0608} = \sin(\lambda r) + \frac{-r}{J_t}\xi_{24}(\lambda),$$

$$H_{0609} = \frac{-J_L r}{J_t}, H_{0610} = \frac{-J_L r}{J_t},$$

$$H_{0701} = \frac{\xi_{11}(\lambda)}{J_t}, H_{0702} = \frac{\xi_{12}(\lambda)}{J_t},$$

$$H_{0703} = \frac{\xi_{13}(\lambda)}{J_t}, H_{0704} = \frac{\xi_{14}(\lambda)}{J_t},$$

$$H_{0705} = \lambda \sin h(-\lambda r) + \frac{\xi_{21}(\lambda)}{J_t}, H_{0706} = \lambda \cosh(-\lambda r) + \frac{\xi_{22}(\lambda)}{J_t},$$

$$H_{0707} = -\lambda \sin(-\lambda r) + \frac{\xi_{23}(\lambda)}{J_t}, H_{0708} = \lambda \cos(-\lambda r) + \frac{\xi_{24}(\lambda)}{J_t},$$

$$H_{0709} = \frac{J_L - J_t}{J_t}, H_{0710} = \frac{J_L - J_t}{J_t},$$

$$H_{0805} = \lambda^2 \cosh(-\lambda r), H_{0806} = \lambda^2 \sinh(-\lambda r),$$

$$H_{0807} = -\lambda^2 \cos(-\lambda r), H_{0808} = -\lambda^2 \sin(-\lambda r), H_{0810} = -\frac{K_L}{EI_Z},$$

$$H_{0905} = \lambda^2 \cosh(-\lambda(r+L)), H_{0906} = \lambda^2 \sinh(-\lambda(r+L)),$$

$$H_{0907} = -\lambda^2 \cos(-\lambda(r+L)), H_{0908} = -\lambda^2 \sin(-\lambda(r+L)),$$

$$H_{1005} = \lambda^3 \sinh(-\lambda(r+L)), H_{1006} = \lambda^3 \cosh(-\lambda(r+L)),$$

$$H_{1007} = \lambda^3 \sinh(-\lambda(r+L)), H_{1008} = \lambda^3 \cosh(-\lambda(r+L)).$$

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
