# Peer review of "Modeling and Disturbance Compensation Sliding Mode Control for Solar Array Drive Assembly System"

_aerospace, doi:10.3390/aerospace10060501_

Round 1

Reviewer 1 Report

The paper presents a global modal methods modelling of flexible solar arrays, to be controlled through a solar arrays drive assembly. The paper also presents the control strategy for the SADA motor, which is based on a disturbance compensation sliding mode controller, in order to smooth and suppress vibrations and other induced errors on the solar arrays motion. The paper is well written, and it is clearly showing its point. I  thus suggest to accept it for publication, but after a few revisions are done on the manuscript.

I list below some specific comments:

- the literature review primarily discusses works on flexible modelling, permanent magnet synchronous motor control and sliding mode methods. I would however suggest to extend it with a survey on recent methods to control solar arrays to point the Sun, both with SADA and without SADA. This is to compare the presented results with respect to the current state of the art for spacecraft's solar arrays pointing control. The comparison should be also done in the result section and in the conclusions. Below some suggestions of work discussing the problem:

   - Kristiansen, Bjørn Andreas, Jan Tommy Gravdahl, and Tor Arne Johansen. "Energy optimal attitude control for a solar-powered spacecraft." European Journal of Control 62 (2021): 192-197.

   - Colagrossi, Andrea, and Michèle Lavagna. "A Spacecraft Attitude Determination and Control Algorithm for Solar Arrays Pointing Leveraging Sun Angle and Angular Rates Measurements." Algorithms 15.2 (2022): 29.

   - Zhang, Jinlong, Shaobo Lu, and Luyi Zhao. "Modeling and disturbance suppression for spacecraft solar array systems subject to drive fluctuation." Aerospace Science and Technology 108 (2021): 106398.

   - Santoni, Fabio, et al. "An orientable solar panel system for nanospacecraft." Acta Astronautica 101 (2014): 120-128.

- the literature review and the paper discussions shall in general be more focused on small satellite applications, given the special issue to which this paper is submitted: Advanced Small Satellite Technology.

 - Parameters in Tab.1 shall be justified. How are they selected? Are they significant for proving the general validity of the present work?  Whic is the dependance of the controller with respect to these parameters?

- Description of Ansys modelling shall be described with more details. Fig.4 is not enough and does not allow to understand the details of Ansys model.

- Fig. 5 requires subtitles and direct mode desciription below the single images.

- Captions of Tab.3 and Tab. 4 may be exchanged. Please, check.

- The selection of the parameters of PI and SM controllers at pag. 13/14 shall be justified and extended. How the controllers have been designed? Which is the sensitivity of the results with respect to these values?

- Fig 7 and Fig 8 do not allow to distingush results between PI, SMC and DCSMC - especially in subfigs (b), (c) and (d). In particular, for black and white visualization of the paper. The authors are encouraged to revise these figures in order to improve readability.

- A sensitivity analysis of the method with respect to other system parameters would be veryu seful. Or, in addition, other results need to be included and critically discussed and compared with respect to the one already in the paper.

- The conclusion should be extended highlighting the robustness of the metod with respect to system variations/uncertainties (see sensitivity analysis) and with respect to other methods for solar arrays pointing.

Reviewer 2 Report

Well written paper on a a significant topic. It would preferable if the results were presented in a more general way so that they could be applicable to structures with a wider range of properties, but nevertheless, the manuscript is acceptable for publication in this reviewer's opinion.

Author Response

Thank you for your suggestions and comments.

Reviewer 3 Report

1. Numerous grammatical and stylistic errors make the paper much more difficult to read. Complete proofreading is necessary.

2. The graphic quality of the drawings must be improved. For example, the descriptions in Figure 1 are too small, and Figure 2 is even illegible. The size of the descriptions should be the same or only slightly smaller than the font in the text. In addition, the images have too low resolution. 

3. Line 162. Section numbers are written in Arabic digits.

4. Line 180. Assumption 4 is not clear - is it about ignoring the effects of cogging and other motor torque disturbances?

5. Line 192. "Govern equation" ?

6. Please explain the meaning of the symbols used in the equations.  Section 2.2 is not very easy to understand, especially when the meaning of the symbols is not clear. If the authors think this analysis is important, please comment on it better.

7. Line 250. "mothed"

8. Throughout the paper, the authors use time derivative symbols, but in equation 37 they introduce the "p" operator. Why?

9. "P_n" and "p_n" symbols are mixed up

10. Equation (40) is wrong. It would be true in the case of an unconnected motor, otherwise the actual moment of inertia should be considered, not just "J_m". Next (43) is ok

11. In equation 45, the asterisk probably indicates a reference signal. If so, you should indicate under what assumptions you can assume that the current regulator is ideal. Especially all delays in the control system are open to question.

12. Lines 415-418.If the method of selecting parameters is not given, it is senseless to compare the results. It is always possible to select PI parameters such that the results are worse than another controller. So how were the PI parameters and SMC parameters selected? What were the selection criteria ?

13. Were the torques shown in Figures 7,8 measured? What was the current waveform? What is the reason for the high level of chattering in the torque signal?

Round 2

Reviewer 1 Report

All my previous comments have been satisfactorily answered.

Author Response

Thank you for your comments and suggestions.

Reviewer 3 Report

The paper may be published after minor changes.

Notes:
1. Table 5 contains the parameters of the controllers, there is no single information on HOW they were selected. If this is the result of a trial-and-error method, it should also be clearly stated.

2. In the summary, exact numerical values are given, describing the advantages of the proposed structure over PID and SMC. However, it is written in an unauthorized generalized form. It should be written that for the analyzed case and the taken parameters of the controller such results were obtained. 
